# High-Efficiency Ion Enrichment inside Ultra-Short Carbon Nanotubes

**DOI:** 10.3390/nano12193528

**Published:** 2022-10-09

**Authors:** Yu Qiang, Xueliang Wang, Zhemian Ying, Yuying Zhou, Renduo Liu, Siyan Gao, Long Yan

**Affiliations:** 1School of Physics, East China University of Science and Technology, Shanghai 200237, China; 2School of Material Science and Engineering, East China University of Science and Technology, Shanghai 200237, China; 3Zhejiang Engineering Research Center for Tissue Repair Materials, Wenzhou Institute, University of Chinese Academy of Sciences, Wenzhou 325000, China; 4School of Physical Sciences, University of Chinese Academy of Sciences, Beijing 100049, China; 5Shanghai Institute of Applied Physics, Chinese Academy of Sciences, Shanghai 201800, China; 6University of Chinese Academy of Sciences, Beijing 100049, China

**Keywords:** carbon nanotube, filling capacity, hydrated-cation–π interaction, ion enrichment

## Abstract

The ion-enrichment inside carbon nanotubes (CNTs) offers the possibility of applications in water purification, ion batteries, memory devices, supercapacitors, field emission and functional hybrid nanostructures. However, the low filling capacity of CNTs in salt solutions due to end caps and blockages remains a barrier to the practical use of such applications. In this study, we fabricated ultra-short CNTs that were free from end caps and blockages using ball milling and acid pickling. We then compared their ion-enrichment capacity with that of long CNTs. The results showed that the ion-enrichment capacity of ultra-short CNTs was much higher than that of long CNTs. Furthermore, a broad range of ions could be enriched in the ultra-short CNTs including alkali-metal ions (e.g., K^+^), alkaline-earth-metal ions (e.g., Ca^2+^) and heavy-metal ions (e.g., Pb^2+^). The ultra-short CNTs were much more unobstructed than the raw long CNTs, which was due to the increased orifice number per unit mass of CNTs and the decreased difficulty in removing the blockages in the middle section inside the CNTs. Under the hydrated-cation–π interactions, the ultra-short CNTs with few end caps and blockages could highly efficiently enrich ions.

## 1. Introduction

Carbon nanotubes (CNTs) exhibit remarkable structural and physicochemical properties [1,2] and show potential in water purification systems [3], ion batteries [4,5], memory devices [6], supercapacitors [7,8] and field emission [9,10,11]. The hollow cavities of CNTs can host guest molecules/ions [12,13,14,15] and confine the guest species in radial directions, effectively forming one-dimensional nanowires or nanoparticle materials [16,17,18]. Furthermore, such host–guest conformations alter and/or enhance the magnetic [19,20], electrochemical [21], optical [22,23], electromagnetic [24,25], catalytic [26,27] and other physicochemical properties [14,18,28] of both the host CNTs and the guest species [29]. Several studies have demonstrated that filled-CNT-based devices offer access to a range of next-generation technologies [13]. However, their realization and commercialization are hindered by an incomplete understanding of the CNT-filling mechanisms and underdeveloped filling technology [13].

The methods for filling CNTs can be categorized as in situ or ex situ [30]. The main in situ methods are arc discharge and catalytic chemical vapor deposition. However, both of these methods generally require complex operation processes and suffer from serious limitations [30] associated with small metal domains. Compounds that do not catalyze the growth of CNTs are hard to be filled in situ in the nanotubes [13]. In addition, in situ filling methods remain unfeasible for filling single-walled carbon nanotubes (SWCNTs) and double-walled carbon nanotubes (DWCNTs) [31,32,33]. Ex situ filling methods have relatively simple operation processes and are suitable for a large range of filling species, thus mitigating the limitations of in situ filling methods [13,30]. There are two common ex situ filling methods. The first is a physical method called ‘capillary-induced filling’ developed by Ajayan and Iijjima [34], which achieves filling by annealing a precursor at a temperature above its melting point. The second is a wet chemical method developed by Tsang et al. [35], which utilizes acidic treatment to open the caps of CNTs, allowing entry to a precursor solution, which may then be transformed chemically by heating or reduction [35,36]. Of these methods described above, the wet chemical method is the most widely used in scientific research due to its relatively simple operation and due to it having the largest filling material range [13,14].

Unfortunately, the wet chemical methods often exhibit low filling efficiency. For example, D. Ugarte et al. point out that although the opening procedures have been optimized, the metal nitrate filling efficiency into their opened long tubes is rather low (2–3%) [36]. Most of the filling materials exist as particles or short rods and distribute periodically far away from the next one along the nanotube [24,25,37], indicating that the filling efficiency is low. As a kind of ex situ method, filling CNTs using the wet chemical method generally involves two steps: (1) opening the closed end caps, and (2) filling the CNTs [30]. Conventional wisdom holds the opinion that a good opening is the precondition of solutions entering CNTs, and many methods to open CNTs have been developed, such as gas-phase oxidation [38,39] and liquid-phase oxidation [36,40,41]. However, these methods still often exhibit very low filling efficiencies, providing CNTs that can be partially or even completely empty [13,42]. To date, the CNT-filling efficiency for a large ion range using wet chemical methods cannot be satisfactory based on these current opening strategies [36,37,39].

In addition to end caps, blockages from the mid-sections of CNTs can occur, which are more difficult to remove. Removing blockages is a significant problem that needs to be resolved, but it has not been discussed in detail in previous works [13,14,30]. CNT blockages can be residual catalyst materials from CNT fabrication [43], amorphous graphite generated during the treatment of CNTs, internal defects, or accumulated filling materials [42]. These blockages hinder the movement of solutions inside CNTs, and result in part of the CNTs among the blockages being empty. It stands to reason that a blockage in the mid-section of a CNT becomes increasingly difficult to remove completely as the length of the CNT increases. Thus, shortening CNTs and then removing blockages is a simple but very useful strategy to fabricate unobstructed CNTs and can further enhance the filling capacity of CNTs. Accordingly, in the present work, ultra-short CNTs were prepared by ball milling, and the blockages inside the CNTs were efficiently removed using acid pickling. Ball milling has been reported to be a high-efficiency method to modify the length, particle size distribution, hydrogen adsorption, specific surface area and dispersion capacity of CNTs [43,44,45]. Acid pickling is a common method used to efficiently remove amorphous graphite, metals and other impurities inside CNTs [13,14,39]. Our previous work demonstrated a useful wet chemical method to fill CNTs, where ions can accumulate inside wide CNTs immersed in very dilute salt solutions [42]. Therefore, the ion-enrichment capacity of ultra-short CNTs soaked in dilute salt solutions was investigated using transmission electron microscopy (TEM) and energy-dispersive X-ray spectroscopy (EDS).

## 2. Materials and Methods

### 2.1. Preparation of Ultra-Short CNTs

Raw long multi-walled carbon nanotubes (MWCNTs) with an average diameter of ~6 nm and an average length of 20 µm, and another kind of MWCNTs with diameters of 20–50 nm and an average length of 20 µm were purchased from Beijing DK Nano Technology Co., Ltd. These two kinds of MWCNTs were sonicated in a 3:1 (*v*/*v*) mixture of 98% H_2_SO_4_ and 70% HNO_3_ for 2 h to open the mouths. Then, the primary processed CNTs were ball milled with a planetary ball mill (QM-3SP04, 450 rpm) for 6–24 h. Finally, the ball-milled short CNTs were again sonicated in a 3:1 (*v*/*v*) mixture of 98% H_2_SO_4_ and 70% HNO_3_ for 40 min to remove nanoscale impurities (residual catalyst, amorphous graphite, internal defects, etc.). Here, sonication using a 100 W ultrasonic cleaner at 70 kHz was conducted. After sufficient washing, centrifugation (10,000× *g*, 3 min) and vacuum filtration, the resulting CNTs were dried at 60 °C for 24 h in a vacuum oven.

### 2.2. CNT Filling Experiments

KCl (purity ≥ 99.5 wt %), CaCl_2_ (purity ≥ 96.0 wt %) and FeCl_3_ (purity ≥ 97.0 wt %) were purchased from Sinopharm Chemical Reagent Corporation (Shanghai, China). The ultra-short CNT samples (with diameters of 20–50 nm) treated by 24 h ball milling and acid pickling were soaked in 0.14 mol/L (M) KCl, CaCl_2_ or PbCl_2_ solution for 20 min with sonication using a 100 W ultrasonic cleaner at 70 kHz. The ultra-short CNT samples were removed from the solution and dried under an infrared lamp for 15–20 min. During the drying process, the temperature was maintained in the range of 30–40 °C.

### 2.3. Characterization

The CNTs were characterized by TEM (FEI Tecnai G2 F20 TEM, 200 kV), and elemental analysis was performed using EDS. Additionally, zeta potential (Malvern Zetamaster, Malvern Instrument Ltd., Malvern, UK) measurement was used to determine the dispersibility of MWCNT nanofluids prepared using ball milling for different times and acid pickling. The zeta potential of the MWCNT nanofluids was measured at pH 7.0.

## 3. Results and Discussion

### 3.1. Shortening of CNTs

Figure 1 shows the schematic of the fabrication of the ultra-short CNTs. The raw long CNTs were shortened by ball milling and the blockages in the mid-section of CNTs were removed using the acid pickling method.

Ball milling has been reported to be an effective method for grinding CNTs to shorten their length [43,44,45]. Two kinds of CNTs with similar initial lengths but different diameters were ball milled and acid pickled. Figure 2a,b display the TEM images of the MWCNTs with an average diameter of ~6 nm before and after the 24 h ball milling treatment. Figure 2c,d show the MWCNTs with diameters in the range 20–50 nm before and after the 24 h ball milling treatment. The 20–50 nm-wide MWCNTs were shorter and had more uniform lengths than the 6 nm-wide MWCNTs. These results indicate that the wider CNTs were more easily and efficiently shortened by ball milling.

The effects of the different ball milling durations on the lengths of CNTs were also investigated. For each kind of the CNTs that were ball milled using a certain duration, the lengths of about 50 random CNTs were measured. These CNTs had lengths ranging from 0.05 to 1.1 µm, therefore, the numbers of CNTs with different lengths were counted at intervals of 0.1 µm. Figure 2e shows the statistical analysis results of the 6 nm-wide MWCNTs and the 20–50 nm-wide MWCNTs. Increasing the ball-milling duration from 6 to 24 h resulted in an increase in the proportion of shorter CNTs. Figure 2f shows that the average lengths of both kinds of MWCNTs decreased significantly with the ball milling duration. The wider MWCNTs showed more effective truncation than the thinner MWCNTs, which was consistent with the results shown in Figure 2a–e.

Acid pickling is a common method used to remove the impurities inside CNTs [13,14,39]. Figure 3a–c shows the TEM images of three kinds of common blockages in mid-section of the raw long CNTs, namely residual catalyst materials, amorphous graphite and internal defects. After the ball milling and acid picking treatment, these blockages were efficiently removed. As shown in Figure 3d, the ultra-short CNTs were unobstructed.

The efficiency of preparing the ultra-short CNTs can be described using the mass ratios of M_short_: M_raw_, which is equal to the mass of ultra-short nanotubes divided by the mass of raw nanotubes. The M_short_: M_raw_ for the 6 nm-wide MWCNTs and the 20–50 nm-wide MWCNTs were 0.68 and 0.80, respectively (Appendix A). The mass losses were ascribed to the removed impurities and some ultra-short CNTs lost during the cleaning processes. If the mass loss during the cleaning processes can be reduced in the future, the applicability of this study will be strong.

Figure 4 shows that the zeta potentials of the two kinds of MWCNTs decreased with the ball milling duration. For the same ball milling time, the wider MWCNTs had a lower zeta potential than the thinner MWCNTs, which can be attributed to the thicker MWCNTs having more negatively charged functional groups on their side walls and ends than the thinner MWCNTs [44,46]. The two kinds of ultra-short CNTs subjected to ball milling for 24 h were dispersed in deionized water using ultrasonic treatment to form suspensions with a concentration of 0.04 wt %, as shown in the insert of Figure 4. No precipitations were observed in the suspensions upon standing for 100 days. However, the raw long CNTs at the same concentration aggregated within one hour (Appendix A). Zeta potentials give an indication of the potential stability of a system, where high zeta potentials (either negative or positive) indicate electrically stabilized particles, while colloids with low zeta potentials tend to coagulate or flocculate [46]. For the purposes of this study, a better CNT dispersion was considered to benefit their filling.

### 3.2. KCl Enrichment inside Ultra-Short CNTs

Our previous work [42] demonstrated the formation of highly concentrated ion aggregations inside MWCNTs soaked in dilute salt solutions. However, the CNT-filling efficiency was always low when the MWCNTs were long, i.e., tens of micrometers in length. Ion aggregations at the mouths of long CNTs were easily observed, as demonstrated by the high-angle annular dark field scanning transmission electron microscopy (HAADF-STEM) image of a CNT soaked in 0.14 M KCl shown in Figure 5a. Except for the tube mouth, most of the long CNT was empty, although some salt aggregations were observed in the middle of the long CNT (Figure 5b).

Accordingly, we reasoned that the unobstructed ultra-short CNTs would show a reduced incidence of blockage by salt aggregation (Figure 5a and Appendix A) or other blocking materials (Figure 3a–c). Figure 5c shows an HAADF-STEM image of an ultra-short MWCNT that was soaked in 0.14 M KCl solution. Salt aggregations filled nearly the entire ultra-short CNT. Elemental mappings showing the distribution of the elements C, O, K and Cl of the salt aggregation in the ultra-short CNT are also illustrated. The signal for C was the same shape as the CNT. The signals for K and Cl were high-intensity and located in the middle of the range of the C signal, and their widths were similar to the inner diameter of the CNT. Furthermore, the O signal, mainly from water inside the tube and functional groups on the tube, had a dispersive width equal to that of the C signal and was a little higher in the middle. Thus, the EDS mapping of the ultra-short CNT demonstrated that the tube was filled with high-concentration KCl solution.

Figure 5d shows the EDS results of the aggregations demonstrating that C, O, K and Cl were present. Furthermore, 16 salt aggregations in ultra-short CNTs were randomly selected and their K/O atomic ratios are shown in Figure 5e, indicating that the concentrations of these salt aggregations might be at least 1–2 orders of magnitude higher than that in the solution outside the CNTs. The K/C mass ratios of the aggregations were also tens of times higher than that in the solution outside the CNTs (see more details in Appendix A). These results demonstrate that the ultra-short CNTs could enrich ions in a dilute solution.

The high concentrations of the salt solutions inside the short CNTs should be attributed to the hydrated-cation–π interactions between the hydrated cations and the π electrons in the aromatic rings of the CNTs [42]. Most carbon-based nanochannel surfaces, such as CNTs, comprise aromatic hexagonal carbon rings rich in π electrons [47]. The non-covalent interactions between a cation and a π-electron-rich carbon-based structure are referred to as cation–π interactions [48]. The polycyclic aromatic ring structure of CNTs is very rich in π electrons, so the cation–π interactions inside are strong enough to cause adsorption of hydrated cations [42].

The confine effect of CNT channels to the solution or water [42,49,50] kept the enriched salts inside the CNTs when the nanotubes left the dilute solutions. The formation of salt blockages further enhanced the stability of these enriched salts inside the CNTs [42]. This indicated that ion adsorption inside the CNTs had certain advantages over that of the outside, due to the fact that the outside surface of raw CNTs without any functional groups usually exhibits weak ion adsorption [51]. We hold the opinion that when the CNTs were moved out from the dilute solutions, water flowed away from the hydrophobic outside surface of the CNTs [52] and took away adsorbed ions outside the nanotubes.

The filling capacity of these ultra-short CNTs can be described in terms of the filling length to total CNT length ratio, i.e., the length of the salt aggregation divided by the length of the corresponding CNT. The length ratios for 16 random ultra-short CNTs were analyzed, and the results are shown in Figure 5f. The highest value was 0.67 and the average for the 16 samples was ~0.49, which is much better than the value of 0.03 for the long CNTs (see more details in Appendix A). Thus, the ultra-short CNTs had a much better filling capacity than that of the long CNTs, as commonly evidenced by the fact that long CNTs filled with guest materials can be partially or even completely empty (Figure 5b) [13,36,37,42].

This highly improved filling capacity of the ultra-short CNTs was ascribed to the decreased incidence of blockage. Figure 6 shows a schematic comparing the filling capacity of raw long CNTs and ultra-short CNTs. The orange and light blue regions in the schematic represent positions that cannot be filled and that can be filled with ion aggregations, respectively. The long CNT is blocked by residual catalyst, amorphous graphite and internal defects, while the ultra-short CNTs have a higher orifice density, shorter length and unobstructed channels. Here, the orifice density of a CNT can be defined as the number of openings per unit mass of CNTs.

Furthermore, our previous work [42] indicated there can be another blockage by ion aggregation when CNTs are soaked in salt solution. Shortening of the length of nanotubes can effectively mitigate the blockage of filling ions as well.

### 3.3. Discussion of Filling Capacity of CNTs

The ultra-short CNTs were less susceptible to blocking, resulting in higher ion-enrichment capacity. We can demonstrate that the ion-enrichment capacity of these ultra-short CNTs is satisfactory through simple mathematics. Let us assume that a CNT comprises concentric zigzag SWCNTs, that the distance between neighboring carbon atoms is 0.14 nm and that the interlayer spacing between the two nearest concentric SWCNTs is 0.34 nm [53]. For a SWCNT of length *l*, the number of carbon atoms in SWCNT layer *k* is:(1)$$n_{k}=(l/0.14)\times (1.33\pi d_{k}/0.14\sqrt{3})$$
where *d_k_* is the diameter of SWCNTs in layer *k*, determined by:(2)$$d_{k}=d+0.68(k-1)$$
where *d* is the inner diameter of the CNT. Thus, the total number of carbon atoms in the CNTs (*n*) is:(3)$$n=\sum_{k=1}^{N}(l/0.14)\times (1.33\pi d_{k}/0.14\sqrt{3})$$
where *N* is the total number of SWCNT layers. The mass of the CNT, *M*, should be:(4)$$M=12n/N_{A}$$
where *N_A_* is the Avogadro’s constant (6.02 × 10^23^/mol). If we assume that the space occupancy of the filling material inside the CNTs is *η*, the mass of the filling material, *m*, is:(5)$$m=0.25\times 10^{-21}\eta \rho \pi d^{2}l$$
where *ρ* is the relative density of the filling material. From the above, we can determine the filling capacity for a specific filling material, *γ*:(6)$$\gamma =m/M=0.456\eta \rho d^{2}/(0.5dN+0.17N(N+1))\;(0\;\leq \;\eta \leq \;1)$$

Thus, the ion-enrichment capacity of a filling material can be assessed conveniently using Equation (6). Figure 7 visualizes Equation (6) when we assume the ultra-short CNTs have a space occupancy of 0.2. The ion-enrichment capacity increases with the inner diameter of CNTs, while it decreases with the increase in the total number of layers. For example, for MWCNTs with an average inner diameter of 7 nm and an average layer number of 20, and assuming the filling material is crystalline KCl with a relative density of 1.98 and a space occupancy of 0.2 (this is just an assumption), the K-enrichment capacity of CNTs is 30.9 mg/g. If the average value of the layer number of MWCNTs, *N*, can be decreased to 3, the K-enrichment capacity of the CNTs increases to ~354 mg/g. Thus, if we develop CNTs with large inner diameters and fewer layers in the future, their ion-enrichment capacity will be satisfactory.

### 3.4. Enrichment of Other Ions by Ultra-Short CNTs

In addition to alkali-metal ion K^+^, the ultra-short CNTs can be used to enrich other ions, for example, heavy-metal ions and alkaline-earth-metal ions.

The high-efficiency enrichment of heavy-metal ions inside ultra-short CNTs will expand the environmental protection values of the nanotubes. Heavy metals are hazardous to the environment owing to their toxicity and carcinogenicity [54,55,56]. Heavy-metal ions cannot be biodegraded, so they persist in the environment and tend to accumulate through the food chain, causing serious problems for human health, plant growth and animal habitats [57,58,59]. Pb is one of the most common and important heavy metals considering its widespread use and high toxicity [60]. Accordingly, the treatment of polluted wastewater containing Pb produced from ammunition production, battery manufacturing, pigment processing and electroplating [61] is a global concern [62]. Adsorption is an efficient, low cost and easily applicable method of conventional wastewater treatment [63]. Therefore, the Pb-enrichment capacity of ultra-short CNTs was evaluated.

Figure 8a shows an HAADF-STEM image of a prepared ultra-short MWCNT that was soaked in a 0.14-M PbCl_2_ solution. Ion aggregations filled most of the ultra-short CNT. Figure 8b shows the EDS mappings of the aggregations. The signals for Pb and Cl were highly intense and located in the middle of the C signal, and their widths were similar to the inner diameter of the CNT. Furthermore, the O signal, mainly from water inside the tube and functional groups on the tube surface, had a dispersive width equal to that of the C signal and was much more intense in the middle. Figure 8c shows the EDS result of the aggregation demonstrating that C, O, Pb and Cl were present. The atomic ratio of Pb to O in the salt aggregation was ~0.25, which was almost 100 times that in the solution outside the CNTs (0.00252). In addition, the Pb/C mass ratios of the aggregations inside the ultra-short CNTs were tens of times higher than that in the solutions outside the CNTs (see more details in Appendix A). Thus, these EDS results demonstrate that the ultra-short CNTs could enrich Pb^2+^ from a dilute PbCl_2_ solution to high concentrations. The length ratios of filling length to total CNTs for 10 random ultra-short CNTs were analyzed, and the results are shown in Figure 8d. The highest value was 0.81 and the average value for the 10 samples was ~0.54, indicating the ultra-short CNTs could efficiently enrich Pb^2+^ ions.

In addition to the heavy-metal ion Pb^2+^, the ultra-short CNTs could enrich alkaline-earth-metal ions such as Ca^2+^ (Appendix A). The salt enrichment inside ultra-short CNTs was attributed to hydrated-cation–π interactions. The strong cation–π interactions between other cations (including Li^+^, Mg^2+^, Cu^2+^, Cd^2+^, Cr^2+^, Ag^+^ and Rh^3+^) and aromatic-ring structures observed elsewhere [47,48] suggest that ultra-short CNTs can be applied to high-efficiency ion accumulation for a wide range of ions.

## 4. Conclusions

The wet chemical method to fill CNTs is widely used in scientific research due to it having a relatively simple operation and a large filling material range [13,14]. However, low efficiencies restrict the applications of the wet chemical method in many cases [13,24,26] and constitute a barrier to the realization and commercialization of certain CNT applications [13]. In the present study, we shortened long CNTs using ball milling and then removed the blockages inside these nanotubes with acid pickling. Then the ion-enrichment capacity of these obtained ultra-short CNTs was compared with that of long CNTs. The results showed that the enrichment capacity of the ultra-short CNTs for ions was much higher than that of the long CNTs due to the ultra-short CNTs being more unobstructed. Alkali-metal ions (e.g., K^+^), alkaline-earth-metal ions (e.g., Ca^2+^) and heavy-metal ions (e.g., Pb^2+^) could be enriched in the ultra-short CNTs. The ion enrichment inside the ultra-short CNTs was attributed to hydrated-cation–π interactions. The strong cation–π interactions between other cations (including Li^+^, Mg^2+^, Cu^2+^, Cd^2+^, Cr^2+^, Ag^+^, and Rh^3+^) and aromatic-ring structures observed elsewhere [47,48] suggest that ultra-short CNTs can be applied to ion accumulation for a wide range of ions. Thus, this study provides valuable information for the application of CNTs to next-generation water purification systems [3], ion batteries [4,5], memory devices [6], supercapacitors [7,8] and field emissions [9,10,11].

## Figures and Tables

**Figure 1 nanomaterials-12-03528-f001:**
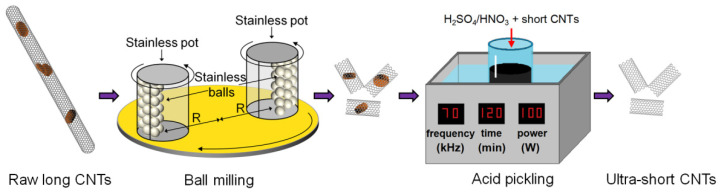
Schematic of the fabrication of ultra-short carbon nanotubes (CNTs) using ball milling and acid pickling methods.

**Figure 2 nanomaterials-12-03528-f002:**
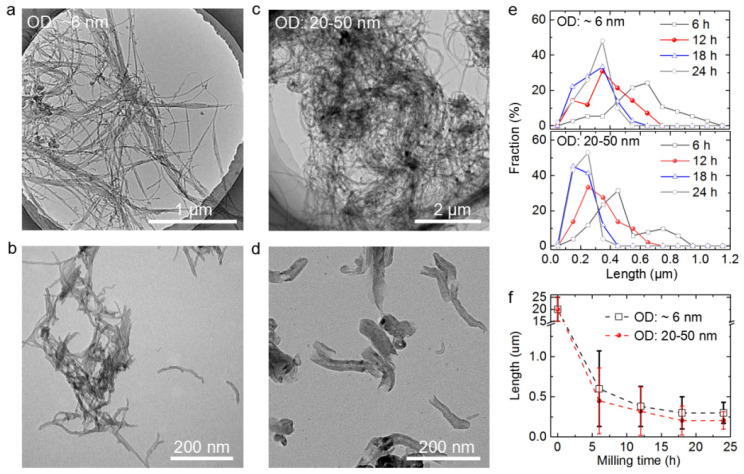
Ball milling treatment results of two kinds of CNTs. Transmission electron microscopy (TEM) images of the 6 nm-wide multi-walled carbon nanotubes (MWCNTs) before (**a**) and after (**b**) 24 h ball milling treatment and acid pickling. Figures (**c**) and (**d**) display the TEM images of 20–50 nm-wide MWCNTs before and after 24 h ball milling treatment and acid pickling, respectively. The upper and lower images in (**e**) display the length statistical analysis results of the 6 nm-wide MWCNTs and the 20–50 nm-wide MWCNTs, respectively. (**f**) Average lengths of both kinds of MWCNTs plotted against ball milling time.

**Figure 3 nanomaterials-12-03528-f003:**
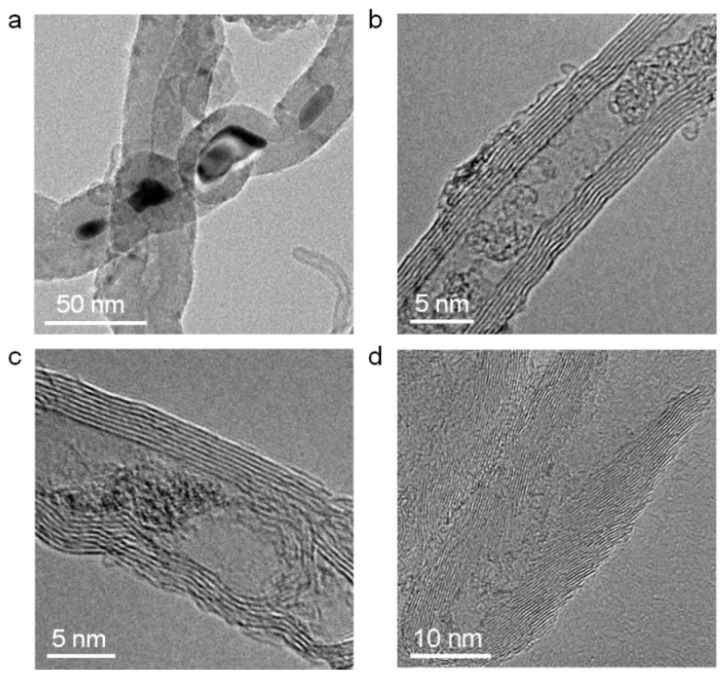
(**a**–**c**) TEM images of residual catalyst, amorphous graphite and an internal defect in the mid-section of CNTs are displayed, respectively. (**d**) A TEM image of an ultra-short CNT.

**Figure 4 nanomaterials-12-03528-f004:**
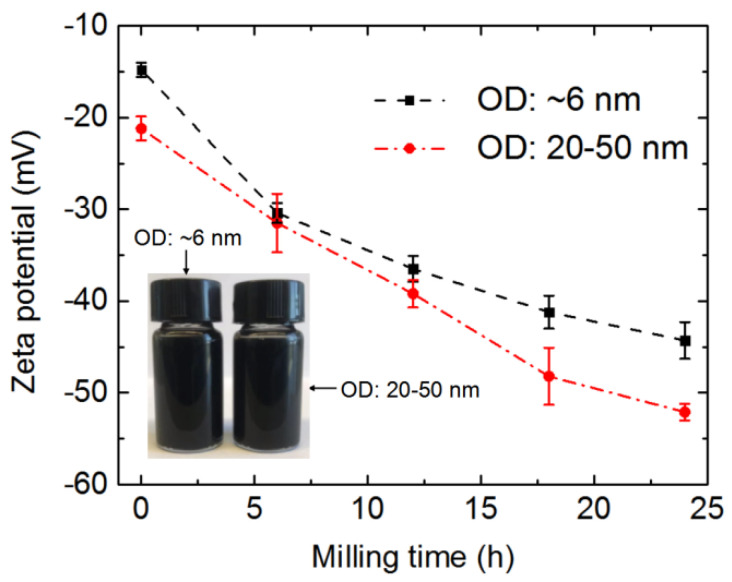
Zeta potentials of two kinds of MWCNTs plotted against ball milling time. Insert photo shows suspensions of two kinds of ultra-short CNTs, which were prepared by 24 h ball milling treatment and acid pickling. These two suspensions were left standing for 100 days.

**Figure 5 nanomaterials-12-03528-f005:**
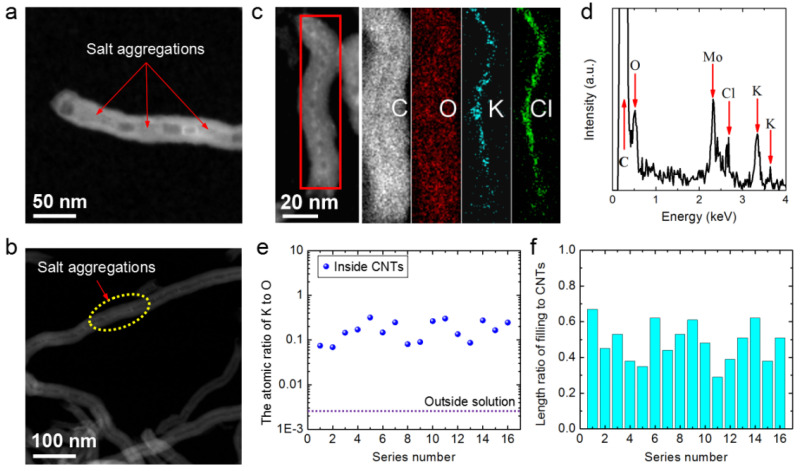
(**a**) A high-angle annular dark field scanning transmission electron microscopy (HAADF-STEM) image of a KCl aggregation at the mouth of a long CNT. (**b**) An HAADF-STEM image showing the mid-section of a long CNT. (**c**) An HAADF-STEM image of a KCl aggregation inside an ultra-short CNT and the corresponding EDS mappings. (**d**) EDS results for the aggregation in (c) (the Mo signal comes from the Mo sample grid). (**e**) K/O atomic ratios (blue points) of 16 aggregations inside 16 random ultra-short CNTs and the K/O ratios of the outside solution (dashed line). (**f**) Filling length/total length ratios for these 16 ultra-short CNTs.

**Figure 6 nanomaterials-12-03528-f006:**
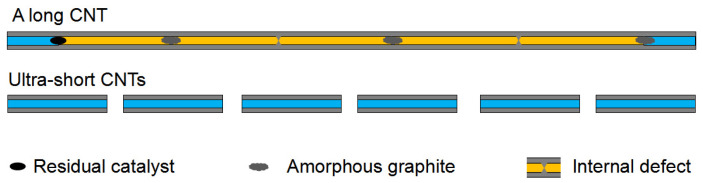
A schematic comparing the filling capacity of raw long CNTs and ultra-short CNTs.

**Figure 7 nanomaterials-12-03528-f007:**
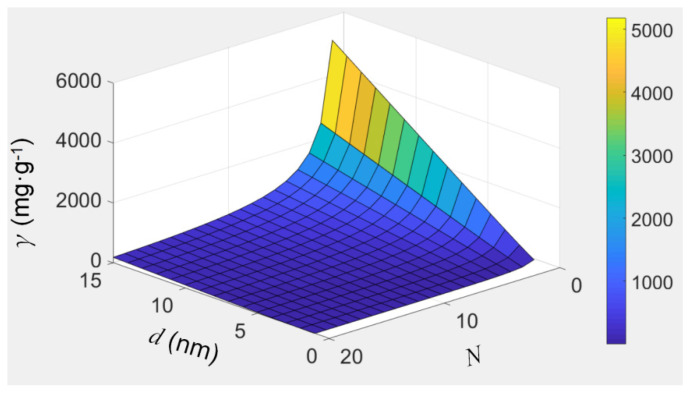
The visualization of equation 6, which shows how the filling capacity of CNTs, *γ*, varies with the inner diameter, *d*, and the total number of layers, *N*.

**Figure 8 nanomaterials-12-03528-f008:**
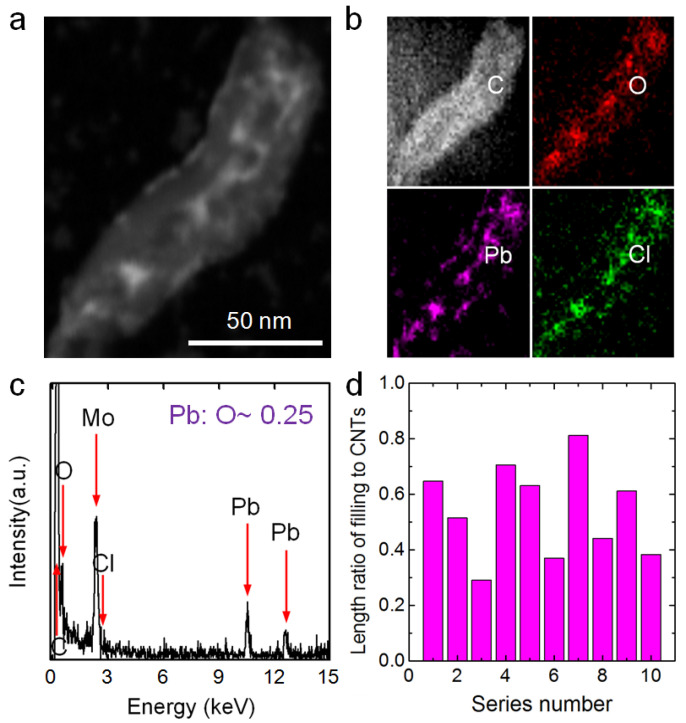
(**a**) An HAADF-STEM image of a PbCl_2_ aggregation inside an ultra-short CNT. (**b**) EDS mappings of elements C, O, Pb and Cl corresponding to the area in (**a**). (**c**) EDS results for the aggregation in (**a**) (the Mo signal comes from the Mo sample grid). (**d**) Filling length/total length ratios for aggregations inside 10 random ultra-short CNTs.

## Data Availability

The authors confirm that the data supporting the findings of this study are available within the article and its Appendix A.

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
