# Peer review of "High-Efficiency Ion Enrichment inside Ultra-Short Carbon Nanotubes"

_nanomaterials, 2022, doi:10.3390/nano12193528_

Round 1

Reviewer 1 Report

The authors described ion enrichment inside ultra-short carbon nanotubes. Their approach is good, but several details are not clear.

2. Materials and Methods. Preparation of ultra-short CNTs. The authors described the preparation of short MWCNTs, but they did not describe the preparation of SWCNTs.

Since they used very harsh methods to open the tubes, they should provide data about the efficiency of their method: mass of open nanotubes / mass of raw nanotubes.

3. Results and Discussion, 3.1. Shortening of CNTs: The SWCNTs are hardly shortened by ball milling. Were they exposed to acid treatment? Since SWCNTs were not used later, they should be removed from the manuscript.

3.2. KCl Enrichment Inside Ultra-short CNTs:

The high concentrations of salt solutions inside short CNTs are attributed to the hydrated-cation–π interactions between the hydrated cations and the π electrons in the aromatic rings of the CNTs. But there are similar aromatic rings on the outside of the CNTs and even on the surface of other carbon nanostructures. Therefore, salt solutions do not need to move inside CNTs. The authors claim that the cation–π interactions inside are strong enough to cause adsorption of hydrated cations, but since the diameter of the tubes is large, there is no significant difference between inside and outside.

The authors should comment this.

3.4. Enrichment of Other Ions by Ultra-short CNTs

The authors used noisy EDS maps to calculate the ratio of Pb to O and ignored the fact that the samples were in UHV. Therefore, their conclusion that the salt concentrations inside the short CNTs are 2 orders of magnitude higher than that of the solutions outside is inaccurate.

They should provide data about mass of ion / mass of carbon ratio, it will give a more accurate picture about the applicability of this method.

I do not think that this material is suitable for applications like drug delivery.

Reviewer 2 Report

Reviewer: Major revision

This paper reports a detailed study about preparing the ultra-short CNTs using ball milling, and removing the blockages inside CNTs efficiently using acid pickling. The proposed approach is very meaningful, and the manuscript organization is satisfied. So, I think that this paper deserves to be published in Nanomaterials after major revision of some issues as follows:

1- Authors must ensure that the quality of English is improved (i.e., make all efforts to rectify any grammatical mistakes, typos, double spaces, missing spaces etc.).

2- The introduction part is suggested to be improved: I) the applications of CNTs are needed to be discussed. The authors must motivate that in the introduction part with one or two sentences and supported them with suitable references, for example: (DOI: https://doi.org/10.3390/molecules26134025), (DOI: 10.1039/D0TC04597G), and (DOI: 10.1039/C9TC03365C).   II) The advantage of ball milling and acid pickling methods are suggested to be discussed in detail. III) The research significance and the difference between this work and previous works are suggested to be improved.

3- FTIR is needed to distinguish between the single-walled and multi-walled carbon nanotubes.

4- It will be better to compare the performance of CNTs in this work with previously reported CNTs based works.

Reviewer 3 Report

 Comments from Reviewer

Title: High-efficiency ion enrichment inside ultra-short carbon nanotubes

The current form's presentation of methods and scientific results is satisfactory for publication in the Nanomaterials journal. The minor and significant drawbacks to be addressed can be specified as follows:

1.                  There are no clearly formulated goals in the reviewed paper.

2.                  Fig. 4, y-axis. Zeta Potential ---> Zeta potential.

3.                  Fig. 6. Blue colour – filled CNT? Pink one – closed space?

4.                  The conclusions are too long.

5.                  Why such a high oxygen content on nanotubes? See Figs.  5c and 8b.

Sincerely,

The reviewer.

Reviewer 4 Report

The authors described ion-enrichment in CNTs, especially the shorter ones with the help of ball milling and sonication in strong acid. The experiments are well-presented and results are intriguing. I hope the authors could address following issues before I recommend this manuscript for publication:

1. First of all, I wonder what the definition of ultra-short is. Is there a specific length or threshold for nanotubes to be ultra-short?

2. Line 226-229, is there a specific number for filling capacity of long CNTs? It would be a much stronger evidence than what the authors have presented in this sentence.

3. Since salt enrichment inside ultra-short CNTs is attributed to hydrated-cation–π interactions, I wonder if the same effect will make the outside of the nanotube more attractive to salt. If not, what is the reason?

4. The reference needs another check. For example, line 123, ref[35] has nothing to do with this manuscript. It seems to me the authors should have cited [33, 34].

Round 2

Reviewer 1 Report

I would like to thank the Authors for the careful revision and detailed answers to my concerns. The missing details were added, the conclusions are better supported and the unsupported claims were removed. The manuscript improved significantly.

Reviewer 2 Report

Authors have fixed the corrections well and the paper now deserves to publish in Journal of Nanomaterials.